# Identifying rheological regimes within pyroclastic density currents

Thomas. J. Jones ®[1] ✉, Abhishek Shetty[2], Caitlin Chalk[3], Josef Dufek[4] & Helge M. Gonnermann[5]

Pyroclastic density currents (PDCs) are the most lethal of all volcanic hazards. An ongoing challenge is to accurately forecast their run-out distance such that effective mitigation strategies can be implemented. Central to this goal is an understanding of the flow mobility—a quantitative rheological model detailing how the high temperature gas-pyroclast mixtures propagate. This is currently unknown, yet critical to accurately forecast the run-out distance. Here, we use a laboratory apparatus to perform rheological measurements on real gas-pyroclast mixtures at dynamic conditions found in concentrated to inter-mediate pumice-rich PDCs. We find their rheology to be non-Newtonian featuring (i) a yield stress where deposition occurs; (ii) shear-thinning behavior that promotes channel formation and local increases in velocity and (iii) shear-thickening behavior that promotes decoupling and potential co-PDC plume formation. We provide a universal regime diagram delineating these behaviors and illustrating how flow can transition between them during transport.

Ground-hugging pyroclastic density currents (PDCs) can occur during all explosive volcanic eruptions and form through a variety of processes[1] such as the collapse of eruption columns[2,3] and lava domes[4], directed blasts[5] and hydromagmatic explosions[6–8]. These currents are less buoyant than the surrounding air, and comprise a multiphase, granular mixture of hot gas and volcanic particles (e.g., pumice, glass, lithics, crystals) that can travel many kilometers away from the vent. Their physical attributes (e.g., particle concentration, grain size, velo-city) are variable over a range of length-scales and timescales, making PDCs among the most complex flows in nature, and as such, they are one of the least understood phenomena of explosive volcanism and remain intensely studied[1,9]. However, given that these widespread gravity-driven currents are responsible for a third of all volcanic fatalities globally[10] and represent the deadliest volcanic hazard, the development of robust hazard models remains a priority for the community[1].

Owing to the lethal nature of these flows, their internal structure remains enigmatic[1,11,12]. Two end members of flow behavior exist, the dilute and concentrated regime end members, continuously spanned by an intermediate regime. The dilute end member is characterised by a low concentration of particles that are held in suspension by the turbulence of the gas phase, whereas in the concentrated end member the volume fraction of particles is much higher, and particles interact by particle-particle collisions, frictional interactions, and pore fluid pressures. A single PDC typically encompasses all regimes both tem-porally and spatially[13–17] and here we focus on the intermediate to concentrated members. Despite major advances in the understanding of PDCs it remains a challenge to simulate these phenomena accu-rately. Due to the bulk behavior of PDCs resembling that of a viscous fluid, the vast majority of PDC models are based on laws of fluid dynamics[9,18]. Currently, there is significant uncertainty pertaining to the shear rheology of the flowing material, and associated constitutive relations used in numerical models. Consequently, this limits the ability of models to accurately predict observed run-out distances without input from empirically modified parameters and thus to reli-ably inform hazard maps and mitigation strategies.

Depth-averaged models have been extensively used to model PDCs[9,11,19], for both concentrated and dilute flows. In the case of depth-averaged concentrated flow models the friction coefficients must be reduced ('tuned') to predict observed runouts[20,21]. One potential and

[1]Lancaster Environment Centre, Lancaster University, Lancaster, UK. [2]Rheology Division, Advanced Technical Center, Anton Paar USA Inc, Ashland, VA, USA. [3]Department of Earth, Ocean and Ecological Sciences, University of Liverpool, Liverpool, UK. [4]Department of Earth Sciences, University of Oregon, Eugene, USA. [5]Department of Earth, Environmental and Planetary Sciences, Rice University, Houston, TX, USA. ✉e-mail: thomas.jones@lancaster.ac.uk

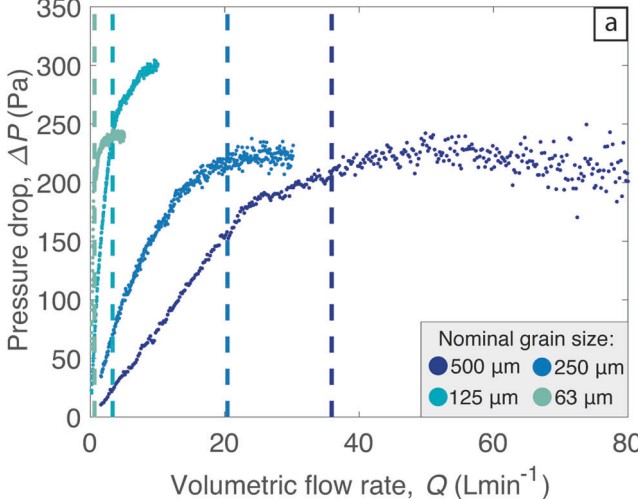

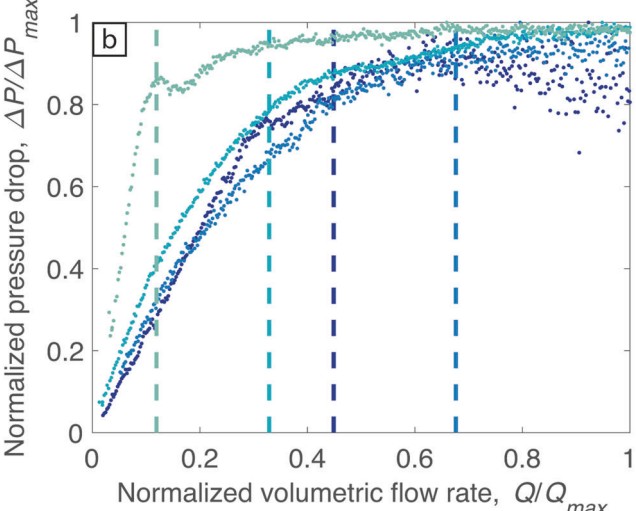

**Fig. 1 | Fluidization behavior of the monodisperse vesicular volcanic particles. a** Pressure drop across the particle bed as a function of the volumetric gas flow rate supplied into the base of the cell. Color symbology is consistent throughout both figure panels. Dashed vertical lines correspond to the $Q_f$ calculated based on particle characteristics. **b** Data in (a) normalised to the maximum values of $\Delta P$ and $Q$ to help visualise the full pressure drop profile for all grain sizes. Note that $Q_{max}$ has no physical meaning, it just represents the maximum flow rate that we used in the pressure drop tests.

often invoked mechanism for friction reduction is fluidization[22]. In a fluidized granular system, some of the normal force and hence friction, is reduced by the flow of escaping gas. Related to this, a common challenge when modeling flow run-out is the inclusion of accurate rheological models. These are constitutive equations that describe material deformation in response to stress during flow and are thus the fundamental building blocks of any flow model. There is debate[20,23,24] about what specific rheology model is appropriate for PDCs. This uncertainty is further compounded by the spatio-temporal variability of the volume fraction of particles in these flows and its impact on the rheology of volcanic gas-particle suspensions[1,25,26]. Current numerical models use rheological equations that are based on simplified (non-volcanic) monodisperse particle systems. The natural case is far more complex than this, with polydisperse gas-particle systems and micro-physical processes that occur on length-scales at the size of an individual particle. Direct studies on the rheology of natural PDC mixtures are sparse[25,27–31]. Wilson and Head[28] deployed a field-based rheometer

to measure the deposits of the 1980 Mt St. Helen's eruption a few days and a few weeks after deposition. These static measurements found the deposits to have a complex, non-Newtonian rheology with yield strengths ranging from 400 Pa to 18,000 Pa and apparent viscosities ranging from 30 Pa s to 13,000 Pa s. Although these data are insightful, they represent the static deposit, and not the dynamic, flowing PDC that they originated from.

Here, we perform a series of scaled laboratory experiments to directly image the internal dynamics and rheological properties of pumice-rich, intermediate to concentrated PDCs, with particle volume fractions, $\phi$ between 0.29 (corresponding to the maximum packing fraction in the unfluidized state, $\phi_m$) and 0.13 (when fluidized). Here we use the scaling of Lube et al.[1] to describe these regimes and emphasize that the transitions between regimes are approximate and based on particle characteristics. Thus, even at the lowest $\phi$ of 0.13, the suspension still falls within the intermediate regime with evident particle clustering. We use a powder flow cell accessory on an air-bearing rheometer that enables direct rheological measurements of variably fluidized, natural pyroclast-gas mixtures at conditions appropriate for PDCs. This allows us to overcome the challenges of direct in-situ measurement. It also enables the systemic variation of flow characteristics (e.g., grain size, fluidization state), whilst simultaneously measuring the dynamic properties (e.g., apparent viscosity). We find that pyroclast-gas suspensions are highly complex and feature different rheological regimes (e.g., shear-thinning, shear-thickening) at specific flow conditions. We encapsulate these data in a regime diagram that allows for the prediction of PDC rheology and mobility based on the eruption characteristics (e.g., fluidization state and shear-stress imposed).

## Results and Discussion
### Fluidization behavior
Two types of experiments were performed using an Anton Paar MCR 302 rotational rheometer, fitted with a powder flow cell (see Methods for full details). A set of pressure drop experiments were used to assess the fluidization state and behavior of the volcanic particles and a set of rotational shear experiments (like conventional rotational rheometry methods[32–38]) measured the internal friction (i.e., torque response) of the granular volcanic material under varying degrees of fluidization. Each experiment focused on a narrow, monodisperse grain size distribution of crushed dacitic pumice particles. We used pumice particles that were separated into 1 mm – 500 μm, 500 μm – 250 μm, 250 μm – 125 μm, and 125 – 63 μm sizes using mesh sieves. They are nominally referred to as 500 μm, 250 μm, 125 μm and 63 μm, respectively. Full grain size distributions obtained by laser diffraction (Fig. S1) show a narrow distribution of the experimental starting material and no particle modification during the experiments due to abrasion or attrition[39,40]. This suggests that the duration of experiments was sufficiently short to avoid significant particle size and shape modifications. Supporting data is shown in Figure S1. We found that the density of the pumice particles, $\rho_p$ varied slightly as a function of grain size (Fig. S2) and we attribute this to the internal vesicle population within the pyroclasts which agrees with previous studies[41].

Upon increasing the air flow rate (also referred to as the flux) $Q$, supplied into the base of the bed of particles, they began to fluidize. This process can be quantified by measuring the pressure drop, $\Delta P$, across the particle bed as a function of $Q$ (Fig. 1; Supplementary Data 1). As the gas flow rate increases, the pressure drop increases approximately linearly until a plateau is reached. Subsequent increases in gas flow rate do not change the pressure drop across the bed of particles. In the fluidized state the bed is observed to bubble. These observations agree with previous volcanological studies that investigated fluidized granular media[22,42,43]. Normalisation to the maximum recorded values of $Q$ and $\Delta P$ shows that this behavior is common to all grain sizes (Fig. 1b). The transition to the plateau in $\Delta P$ marks the onset of

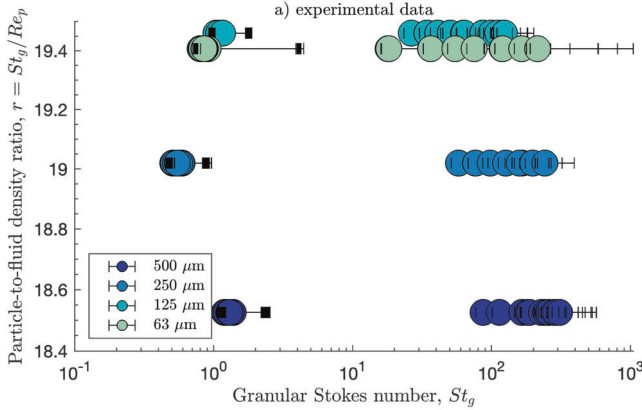

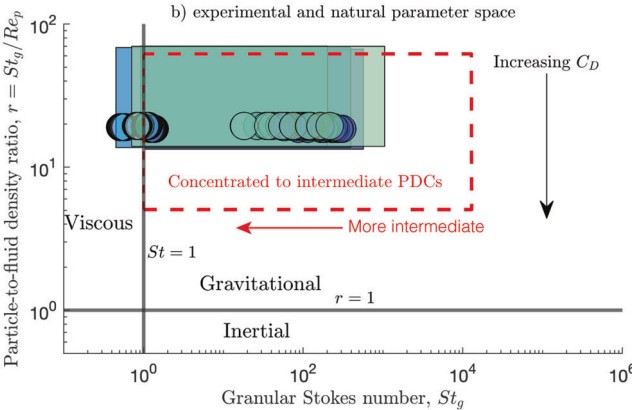

**Fig. 2 | A regime diagram depicting the different behaviors of granular flows. a** The dimensionless numbers for the experimental data, where the error bars show the range covered by varying the representative grain size diameter, $d_p$ from $d_{10} - d_{60}$. The number in the subscript (10 or 60) refers to the percentage volume of the sample volume that is finer. The median particle size $d_{50}$ was used as the representative value. The legend labels correspond to the nominal catching sieve size. A constant drag coefficient of $C_d = 5.2$ was assumed. **b** A regime diagram for natural PDCs with the experimental data overlain. The filled rectangles represent the parameter space of the experimental data when varying the drag coefficient between 0.4 and 10 (see Methods). The dashed red rectangle denotes the parameter space for concentrated to intermediate pyroclastic density currents[1].

---

fluidization, corresponding to a value of $Q = Q_f$. At this gas flow rate the total weight of particles is supported by the up flowing gas, i.e., the drag force exerted by the gas is equal to the buoyant weight of the particles. The corresponding minimum fluidization velocity, $u_f$, and corresponding minimum fluidization gas flux, $Q_f$, can be calculated theoretically[44] (see Methods). We find that $Q_f \approx 35.9$, 20.4, 3.3 and 0.6 L min$^{-1}$ for grain sizes of 500 μm, 250 μm, 125 μm and 63 μm, respectively. These values (shown as dashed vertical lines in Fig. 1) are located close to the start of the $\Delta P$ plateau, and therefore match our experimental dataset well.

Fluidization can also be documented by monitoring the evolution of torque of a 2-blade stirrer immersed in the particle bed, connected to the rheometer head which is rotated at a slow rotation speed (8 revolutions per minute). The torque, $M$ required to rotate the spindle decreases with $Q$ until $M$ plateaus at $Q_f$ and only small torques on the order of 1 μN m or less are required for spindle rotation. This transition equally marks the onset of fluidization[37,38]. During the pressure drop experiments, contemporary visual observations recorded the height of the particle bed, which increased with increasing gas flow rates (Table S1) and were used to calculate the particle volume fraction as a function of gas flow rate, $Q$.

## Dynamic similarity

The dynamic similarity between small-scale laboratory experiments and natural flows is assessed by comparing dimensionless groups of parameters that describe the governing force balances that determine the fluidization dynamics. Several scaling approaches have been used for granular flows, and pyroclastic density currents in particular, to account for particle-particle and particle-fluid interaction. In the case of intermediate to concentrated PDCs, bulk flow dynamics are governed by the behavior of individual grains, and their interaction with each other and the interstitial fluid[1,9]. Depending on the timescale at which an individual particle settles in an interstitial fluid, most natural granular flows (including PDCs) may be in one of three different regimes: viscous, inertial, and gravitational[1,45,46]. These different flow regimes are classified according to three dimensionless numbers: the granular Stokes number $St_g$, the particle Reynolds number $Re_p$, and the ratio of the two – the drag-normalised particle-to-fluid density ratio $r$. For concentrated granular flows these dimensionless numbers are defined as follows[1,45]:

$$St_g = \frac{\alpha \chi d_p \sqrt{\rho_s \sigma}}{\eta_f} \sqrt{\frac{2}{3}} \qquad (1)$$

$$Re_p = \frac{\alpha \chi d_p \sqrt{\frac{2}{3} \sigma \rho_f C_d}}{\eta_f} \qquad (2)$$

$$r = \sqrt{\frac{\rho_s}{\rho_f C_d}} = \frac{St}{Re_p} \qquad (3)$$

Here $\rho_s$ and $\rho_f$ are solid and fluid density respectively, $d_p$ is the representative particle diameter, $\eta_f$ is interstitial fluid viscosity, $\sigma$ is normal stress, $\alpha = k/d_p^2$ is a constant relating the mixture permeability $k$ and particle diameter (see Methods). $C_d$ is the drag coefficient and $\chi, 0 \leq \chi \leq 1$ is the average particle sphericity. The sphericity is a measure of the similarity to a sphere ($\chi = 1$), and typically lies in the range 0.65-0.92 for fine volcanic ash[43]. The granular Stokes number, $St_g$ (Eq. 1), describes the influence of gravitational and viscous forces on the timescale of a particle settling in a concentrated flow. $St_g$ decreases with both increases in the interstitial fluid viscosity and with reduction in permeability. The ratio of inertial to viscous forces gives the particle Reynolds number, $Re_p$ (Eq. 2), whereas the particle-to-fluid density ratio, $r$ (Eq. 3), represents the ratio of gravitational and inertial forces on particle settling.

In nature, concentrated to intermediate PDCs dominantly lie within the gravitational regime[1], where a high content of fine ash (i.e., small $d_p$), large particle density contrast compared to the carrier fluid, and low carrier fluid viscosity result in particle settling being dominated by gravity, rather than by viscous drag or inertial forces. Gravitational granular flows are characterised by dimensionless numbers of $St_g > 1$ and $r > 1$. Within this regime, the characteristic granular Stokes number of concentrated to intermediate PDCs ranges[1,9] between ~$10^0 - 10^4$, with dilute PDCs typically having $St_g < 1$. Viscous drag is significant for PDCs that fall in the lower end of this range (as $St_g$ approaches unity), resulting in high pore pressures and particles being advected with the fluid flow. Pore fluid pressure decreases with increasing granular Stokes number, and the higher end of the natural range ($St_g \sim 10^4$) represent dry, granular, concentrated PDCs with low pore pressures. The values of $St_g$ and $r$ for our experiments are shown in Fig. 2a. The granular Stokes number ranges from slightly less than 1 (viscous dominated) to the order of $10^3$ (gravity dominated). Under a constant drag coefficient assumption, the particle-to-fluid density ratio, $r$, is similar for all experiments, and only varies with the particle density, $\rho_s$ (Fig. 2a). The lowest $St_g$ values correspond to the experiments with no gas flux ($Q = 0$ L min$^{-1}$), which have a significantly lower

permeability than the other experiments, but we note that pore pressure here remains low due the lack of an additional gas source. In the Supplementary Information we provide a detailed breakdown of the individual terms comprising the granular Stokes number (Fig. S3).

The range of $St_g$ and $r$ for natural concentrated to intermediate PDCs are shown as the red dashed box in Fig. 2b, along with our experimental data. Most of the experiments fall within the regime of natural PDCs (Fig. 2b). The experiments with no gas flux ($Q = 0$ L min$^{-1}$) have the smallest granular Stokes numbers, due to their low mixture permeabilities (see Fig. S3) and resultant large viscous drag (despite the fact that there is no air injection). These data lie at the transition between the gravitational and viscous regimes, slightly outside of the expected natural range for concentrated to intermediate PDCs. However, all our experiments with a non-zero gas flux are within the gravitational regime and are the focus of this study. To test our assumed drag coefficient of $C_d = 5.2$, we also plot the scaling results when using a drag coefficient from $0.4 - 10$, representing the full natural range[47] (shown by the shaded boxes). As the drag coefficient is increased, the particle-to-fluid density ratio decreases and gravity becomes less dominant, yet the experiments do not reach the inertial regime (characterised by low $r$ values) – so no matter what $C_d$ is approximated, our experiments remain in the gravitational regime of

natural PDCs. Full details on how the dimensionless parameters were calculated for the experimental data are provided in the Methods. Our scaling results show that the experimental parameter space overlaps with the range expected for natural PDCs, indicating that the experimental results are both representative of and are dynamically scaled to the natural system.

## The bulk apparent viscosity of a PDC

Rotational rheometry experiments were performed on our gas-particle mixtures to determine their rheological properties in a fluidized state. Equivalent to conventional rheometry measurements[32,48,49], a spindle or measuring geometry is inserted into the bed of volcanic particles and rotated at a series of specified rotation rates whilst the torque is simultaneously recorded. These measurements can then be directly related to the shear-rate imposed and apparent viscosity of the system (see Methods). The novelty of our experiments is the measurement of the apparent shear viscosity of variably fluidized natural volcanic gas-particle mixtures, to provide constraints on the mobility of PDCs.

For all grain sizes and for all gas flow rates the apparent viscosity of the gas-pyroclast suspension depends on the imposed shear-rate. In other words, all pyroclast suspensions, irrespective of their fluidization state, exhibit a non-Newtonian rheology (Fig. 3; Supplementary

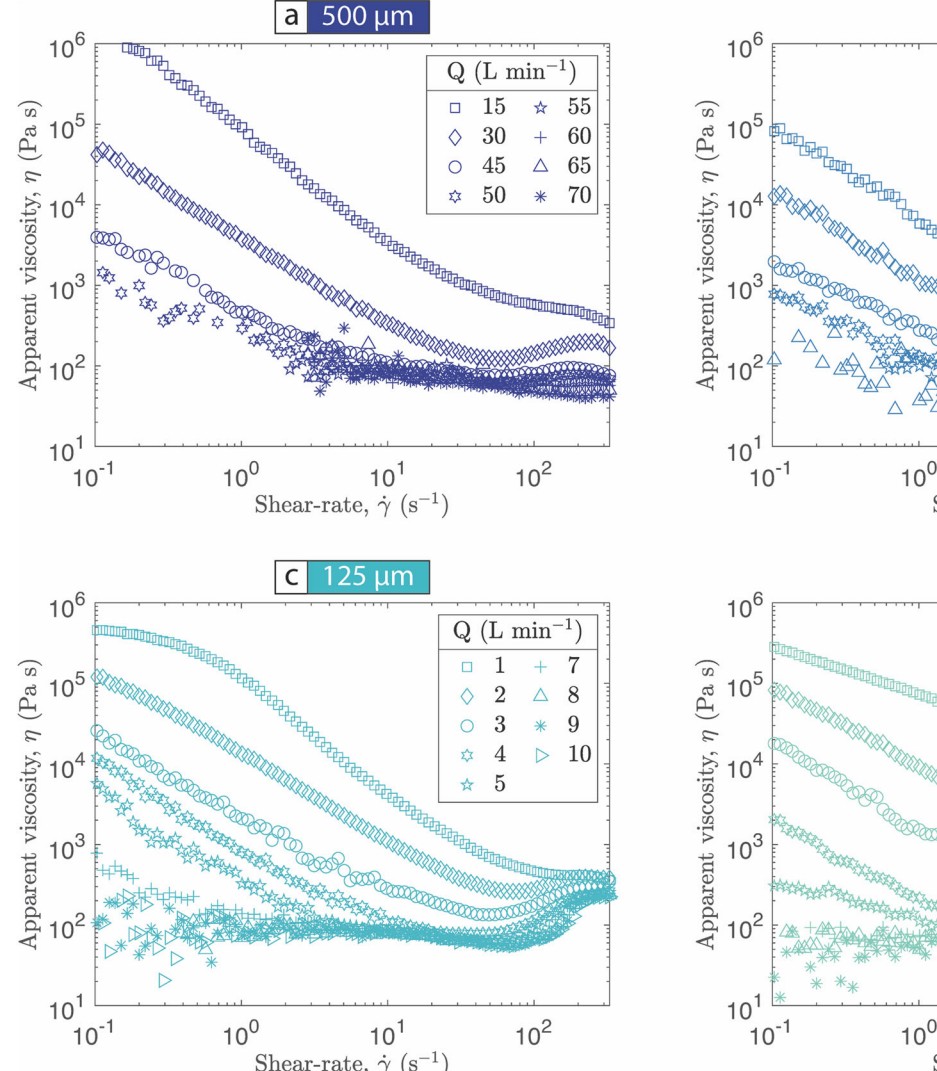

**Fig. 3 | Rheology of gas-pyroclast mixtures scaled to pyroclastic density currents.** The apparent viscosity, $\eta$ (Pa s), of variably fluidized **(a)** 500 µm, **b** 250 µm, **c** 125 µm **(d)** 63 µm pyroclasts as a function of the imposed shear-rate, $\dot{\gamma}$ (s$^{-1}$). Data

point symbols correspond to the gas flow rate, $Q$ (L min$^{-1}$), and are listed within the legend in each figure panel.

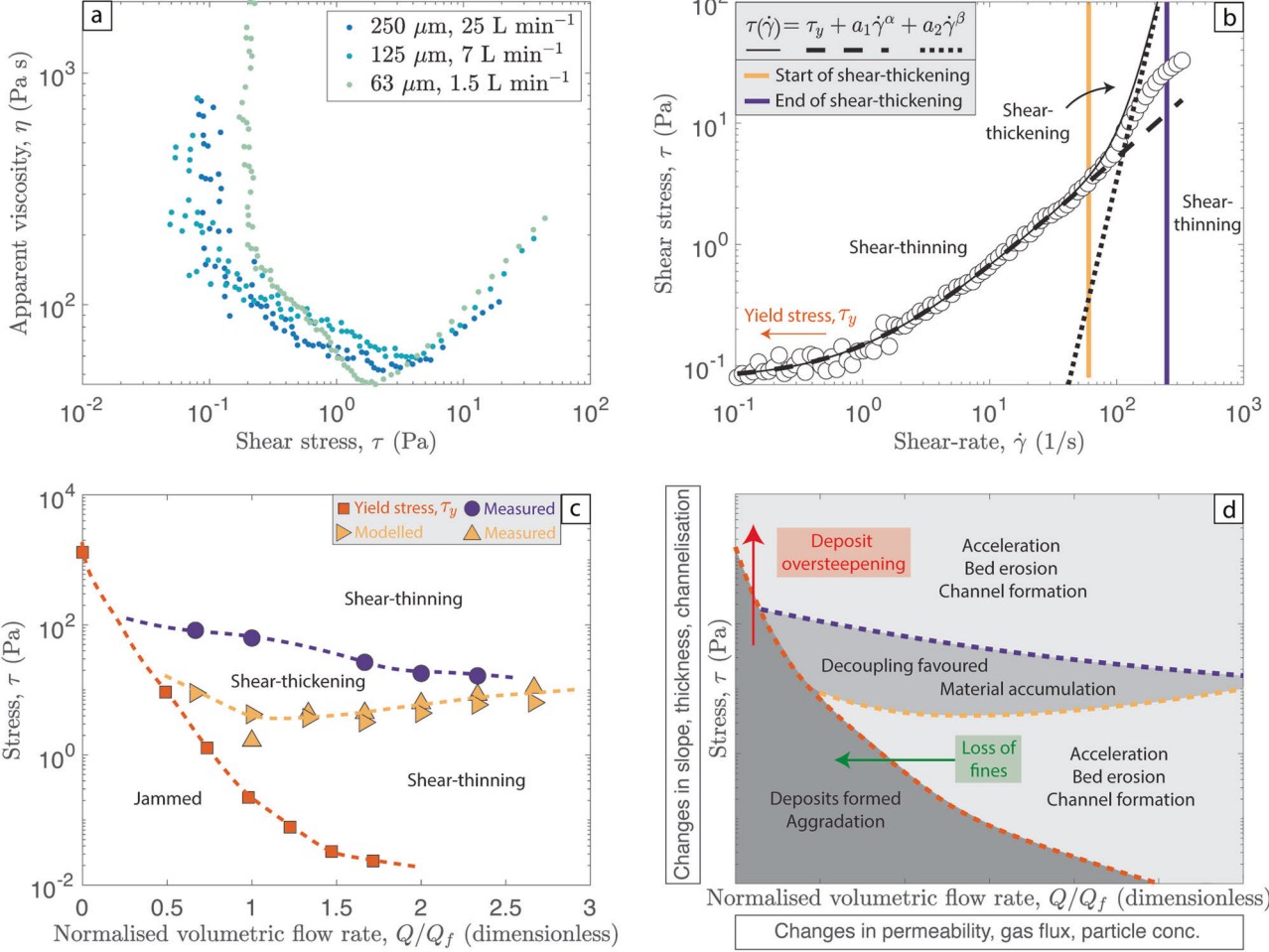

**Fig. 4 | Concentrated to intermediate PDCs as complex, non-Newtonian fluids. a** The apparent viscosity as a function of shear stress for three representative experimental datasets. At low shear stress, at values close to the yield stress the apparent viscosity is extremely high and the suspension becomes jammed and immobile. **b** A fit of the shear stress – shear-rate curve for the representative 250 μm, 25 L min⁻¹ experiment. The white circles represent the data obtained by the rheometry experiments. The black dashed line is the Herschel–Bulkley fit to the lower shear-thinning regime: $\tau(\dot{\gamma}) = \tau_y + a_1\dot{\gamma}^\alpha$. The black dotted line is the fit to the shear-thickening regime: $\tau(\dot{\gamma}) = a_2\dot{\gamma}^\beta$. The solid black line is the sum of both the dashed and dotted line (Eq. 4). This approach was conducted on all experiments, but for illustrative purposes only one dataset is shown. **c** A rheological map of the different rheological regimes shown in our experiments and in PDCs. The orange squares mark the yield stress and the upper limit of the jammed regime where the granular suspension is locked and immobile. The upwards pointing and sideways pointing yellow triangles mark the measured and modeled lower limit to the shear-thickening regime. The purple circles mark the measured upper limit to the shear-thickening regime. This regime diagram was produced for the 250 μm experiments. The experiments performed using other grain sizes also exhibit the same rheological regimes (Fig. S4). **d** Implications of the rheological regime diagram for natural PDCs. Arrows denote how migration between regimes might occur.

Data 2). At low shear-rates the highest apparent viscosities are observed and are on the order of 10⁶ to 10⁵ Pa s. With increasing shear-rate the apparent viscosity of the pyroclast mixture typically decreases to a minimum of ≲10² Pa s, corresponding to a critical shear-rate, $\dot{\gamma}_{c1}$, above which the apparent viscosity increases with shear-rate. These two modes of rheological behavior are termed shear-thinning and shear-thickening respectively. For some experiments, corresponding to intermediate gas flow rates, a second transition is observed where the suspension returns to shear-thinning behavior at higher shear-rates.

For any given grain size, an increase in the gas flow rate, $Q$, lowers the apparent viscosity of the first shear-thinning regime and also suppresses the shear-thinning effect (i.e., the rate in viscosity decrease as a function of shear-rate is lower at higher gas flow rates). Furthermore, in some cases (e.g., $d = 125$ μm, Q = 9 and 10 L min⁻¹; Fig. 3c) the shear-thinning behavior is sufficiently reduced for the suspension to effectively behave as a Newtonian fluid over a restricted shear-rate range. For all grain sizes at sufficiently high gas flow rates the rheological response becomes similar – curves in Fig. 3 become closer together.

## The role of yield stress – a rheological cut-off

Our apparent viscosity data can also be analyzed as a function of the shear stress (Fig. 4a), and we explore the relationship between the shear-rate and shear stress response during flow (Fig. 4b). Typically, the apparent viscosity rapidly increases and tends to infinity below a critical shear stress. This is the manifestation of the yield stress – a critical stress, $\tau_y$, that must be exceeded for flow to occur[50,51]. The origin of this yield stress can be wide ranging and typically occurs through particle-particle interactions that generate a network[52–54] (e.g., jamming, attraction, friction). We find that for all grain sizes tested the yield stress reduces as the gas flux increases and, in general, the yield stress is lower for smaller particle sizes. Even at gas flow rates above complete fluidization (e.g., double the minimum fluidization gas flux; $Q_f$) small yield stresses of ~2 Pa are recorded. The implications of a yield stress, both above and below $Q_f$, are wide ranging and apply to both PDCs in motion and to static deposits.

The presence of a yield stress can be one explanation for the formation of steep sided levees and lobes on flow margins[28,55,56], and it can provide evidence for the onset of deposition and progressive

aggradation within vertically stratified flows. While changes in grain sizes due to sorting may further accentuate these features (in particular levee formation), previous work on dry granular flows with narrow grain size distributions has still exhibited self-channelization and levee formation owing to changes in rheology[57]. When the shear stress at the base of a current fails to exceed the yield stress, we expect deposition to initiate, or in some cases the mixture could travel as a plug flow. Given that the grain size distributions within a current are typically vertically stratified, the critical yield stress required for deposition is not homogenous. For example, both the fining upward profile and upward reduction in particle volume fraction of a current corresponds to an upward reduction in yield stress required for deposition. Thus, successively lower yield strengths will need to be exceeded as the static-flow interface migrates upward (i.e., aggradation) and the current becomes ever finer grained and more dilute with distance from the vent. This can occur up until a dilute concentration limit where particle-particle interactions and any corresponding yield strength are negligible. Such insights support the model of progressive aggradation from the flow base[13,58] and have direct implications for forecasting the run-out distance achieved by PDCs.

Furthermore, many depth averaged models implement a retarding stress to account for the interaction with the substrate[18]. This retarding stress can be crudely compared to the yield stress computed here, with the acknowledgement that many granular materials experience hysteresis where yielding (overcoming the yield stress for flow to occur) and jamming (flow stagnation) can occur at different stresses[59]. A common example is that the angle of repose of avalanches can differ from the angle at which they arrest. The physical interpretation is perhaps most intuitive in the case where Coulomb frictional retarding stresses are used in the depth averaged models. In that case, an apparent friction coefficient could be computed by tracing the interface of the jammed region (i.e., suspension conditions at, or below the yield stress) as a function of gas flux/fluidization state. Other retarding stresses have been used based on other physical models or empirical relationships. For example, empirical constant retarding stresses have been used to match the runout conditions in debris avalanches and pyroclastic density currents[20]. In these cases, the empirical fitting constant could be computed for a specific gas flux/fluidization state.

## A regime diagram for the rheological properties of PDCs

Our experiments show that at shear-rates beyond those required to exceed the yield stress, a region of shear-thinning behavior occurs, followed by a region of shear-thickening behavior at even higher shear rates (cf. Figure 3). These relations can be described using a modified Herschel–Bulkley equation[54,60]:

$$\tau(\dot{\gamma}) = \tau_y + a_1\dot{\gamma}^{\alpha} + a_2\dot{\gamma}^{\beta} \qquad (4)$$

where $\tau_y$ is the yield stress, and $a_1, a_2, \alpha$ and $\beta$ are fitting factors derived empirically. The second term represents the shear-thinning stress ($0 < \alpha < 1$), and the third term represents the shear-thickening stress (with $\beta > 1$). As shown by Fig. 4b, Eq. 4 fits our data well. The suspension first transitions from shear-thinning to shear-thickening at a critical shear-rate value, $\dot{\gamma}_{c1}$, corresponding to the local minima of the viscosity-strain rate curve (when $\frac{\partial \eta}{\partial \dot{\gamma}} = 0$). We differentiate Eq. 4 to eliminate parameter $a_2$, and then evaluate Eq. 4 at $\dot{\gamma}_{c1}$ to find an expression for the critical shear stress, $\tau_{c1}$, as a function of the critical shear-rate:

$$\tau_{c1} = \tau_y + a_1\dot{\gamma}_{c1}^{\alpha} + \frac{1}{2\beta - 2}\left(\tau_y + a_1\dot{\gamma}_{c1}^{\alpha} + \tau_y\right) \qquad (5)$$

The critical shear stress predicted by the model (Eq. 5) closely coincides with the measured experimental values of $\dot{\gamma}_{c1}$. These results can be encapsulated within a regime diagram (Fig. 4c; Fig. S4) in shear stress – volumetric flow rate space. At shear stresses below the yield stress the mixture is jammed and immobile. With increasing the volumetric gas flow rate (i.e., increasing the fluidization state of the gas-pyroclast mixture) the jammed region decreases as the yield stress is reduced. At stresses above the yield stress a large region of shear-thinning behavior exists with a smaller lens of shear-thickening behavior located within (Fig. 4c).

In natural PDCs these various rheological behaviors would manifest in different ways (Fig. 4d) and have a range of possible implications. When the gas-pyroclast mixture is jammed, aggradation would occur, forming deposits, as the yield stress is no longer exceeded. In the shear-thinning regime, flow acceleration, the localization of flow into channels and bed erosion would all be expected. Whereas in the shear-thickening regime material accumulation would occur and decoupling within the current is favored. Such decoupling may promote the segregation of fines from the current top and form co-PDC plumes, provided there is availability of buoyant, hot gas. Different PDCs, and even spatial and temporal variations within the same flow, are likely to occupy different parts of this regime diagram. Changes in the substrate gradient (i.e., slope), the flow thickness, and the formation of channels (i.e., flow focussing) will all act to change the local shear stress within the gas-pyroclast mixture. Changes in the permeability structure within the PDC will change the fluidization state and thus, the normalised volumetric flow rate (Fig. 4d). For example, the elutriation of fines may cause the PDC to enter the jammed regime and stop flowing (green arrow; Fig. 4d); or a deposit might become over steepened and exceed the yield stress required for flow (red arrow; Fig. 4d). For a given set of conditions, reducing the particle volume fraction (e.g., from concentrated to intermediate PDCs) could be conceptualised as increasing the normalized volumetric gas flow rate on this regime diagram (Fig. 4d). While we intentionally focused on relatively simple granular mixtures further work is needed to identify these regimes in broader grain size distributions.

Here, we have performed a set of rheology experiments on variably fluidized gas-pyroclast mixtures. We have shown that these mixtures, scaled analogs of concentrated to intermediate PDCs, are rheologically complex. They exhibit a yield stress even when fluidized and this can help explain deposit, flow, and aggradation characteristics. Non-Newtonian, shear-thinning and shear-thickening regimes have also been identified to occur at specific degrees of fluidization and shear stress. We encapsulate this complex behavior within a regime diagram (Fig. 4) that is valid for all intermediate-concentrated, pumice-rich PDCs worldwide. Furthermore, the concepts from our rheological regime diagram can equally apply to other dense granular systems such as avalanches, aeolian sediment transport and debris flows that fall within the gravitational transport regime.

## Methods

### Input material (Mt. Meager pumice)

The Mount Meager volcanic complex is a calc-alkaline stratovolcano complex situated approximately 150 km north of the city of Vancouver, in southwestern British Columbia and belongs to the northernmost extension of the Cascade Volcanic Arc[61]. The most recent eruption[62] of Mount Meager is dated to 2360 BP and produced explosive and effusive dacite volcanic deposits including: pyroclastic fall deposits, pyroclastic flow deposits, and lava[63]. For this study, large fragments (>10 cm) of pumice were collected from proximal to medial outcroppings of the 2360 BP pyroclastic fall deposit. Samples were collected from fresh deposits within an area of quarrying operations. Any banded or welded pumice fragments were avoided. These pumice blocks were crushed and sieved using a sieve stack with mesh sizes of 1 mm, 500 μm, 250 μm, 125 μm, 63 μm. Throughout this paper we refer to each grain size fraction by its catching sieve, for example 250 μm

input material was caught in the 250 μm mesh, it therefore contains ash fragments >250 μm but <500 μm.

## Particle density

The density of the pumice particles was calculated using an analytical balance to measure mass and a Micrometrics Accupyc II 1340 helium pycnometer to measure volume. A minimum of five aliquots of each nominal grain size was measured ten times for mass and volume, then plotted as mass against volume. A linear regression was then fitted through the data sets and the origin, with the gradients determining the density. Using this method, we determined the particle density to be 2186.2, 2304.3, 2412.3, and 2399.1 kg m$^{-3}$ for the 500 μm, 250 μm, 125 μm and 63 μm grain size fractions respectively. These full datasets are shown in Figure S2.

## Particle size distributions

Each grain size fraction was measured for grain size before and after the rheometry experiments. This was done for two main reasons: (i) to assess if any particle size modification occurred during the rheometry experiments and (ii) to allow us to calculate a weighted average grain size. All samples had their grain size measured using a Malvern Mastersizer 2000 with a water dispersion module attached (capable of measuring particles 0.02 μm to 2000 μm in size.) Using a pump speed of 1900 revolutions per minute a sample aliquot was added to the dispersion module containing deionised water and measured three times. An ultrasonic pulse was applied to the sample for 2 s before the measurement to prevent particles from aggregating in the water suspension. We assumed a particle refractive index of 1.5 and an absorption coefficient of 0.1. For each experimental product three separate aliquots were taken, each measured three times; therefore, the results presented represent averages of nine measurements (Figure S1). Between each sample the entire system was flushed two times to ensure the measurement device was clean and particle free.

## Rheology experiments

All measurements presented here were performed using an Anton Paar MCR302 rotational rheometer with an Anton Paar powder flow cell attached. A schematic of this experimental set up, including the profiled cylindrical measuring geometry is shown in Figure S5. The geometry is 24.16 mm in diameter and contains 20 evenly spaced depressions of 1.75 mm that extend the entire length of the measuring cylinder. The profiled nature of this geometry prevents particle slip during rotation and its combination with the powder flow cell has been proven to be a reliable method to measure the rheology of granular media[64].

In this study two types of experiment were performed: (i) pressure drop experiments to characterise the fluidization behavior and (ii) shear-rate sweeps to characterise the rheological behavior of our pyroclast-gas mixtures. In the pressure drop experiments, a known mass ~ 50 g of pyroclasts, prepared to a restricted grain size range as previously detailed was loaded into the powder flow cell with a 2-blade stirrer measuring geometry inserted. Then using the mass flow controller, the ambient temperature air supplied to the base of the powder flow cell was slowly increased and the pressure drop across the bed was simultaneously measured. Furthermore, upon increasing the volumetric gas flow rate supplied to the bed the 2-blade stirrer was rotated at a constant rate of 8 revolutions per minute and the torque response was recorded. The samples of different grain size required different gas flow rates to fluidize and capture the full range of behavior. The volumetric gas flow rate was increased from 0 L min$^{-1}$ to 80 L min$^{-1}$ for the 500 μm sample, to 30 L min$^{-1}$ for the 250 μm sample, to 10 L min$^{-1}$ for the 125 μm sample, and to 5 L min$^{-1}$ for the 63 μm sample.

For the shear-rate sweep experiments ~50 g of sample was loaded into the powder flow cell with the profiled concentric

cylinder measuring geometry inserted. Then for a constant gas flux the measuring geometry applied a range of shear-rates starting at 0.1 s$^{-1}$ ramping up to 328 s$^{-1}$ with ~20 data points per decade. As in conventional rotational rheometry experiments[48,49,65], for each shear-rate (i.e. rotation-rate) applied the torque was measured and used to determine the (apparent) viscosity of the system. This was done using Anton Paar's RheoCompass software and according[35] to DIN53018-1 and ISO 3219. To ensure that the steady-state (i.e., not start up flow) behavior was measured, the viscosity values were calculated using the final 50% of the imposed rotation. For example, for the lowest shear-rate of 0.1 s$^{-1}$, the spindle was rotated 1260° over ~ 52 s and the viscosity was calculated using the final ~26 s. For the highest shear-rate of 328 s$^{-1}$, the spindle was rotated $1.46 \times 10^5$ degrees over ~0.9995 s and the viscosity was calculated using the final ~0.5 s. If the rare case when no steady state behavior could be achieved, the measurement point was skipped.

These shear-rate sweeps were performed for all grain sizes from 500 μm to 63 μm at a range of volumetric gas flow rates. Specifically for the 500 μm sample the rheology experiments were performed at 0, 15, 30, 45, 50, 55, 60, 65, and 70 L min$^{-1}$. For the 250 μm sample the rheology experiments were performed at 0, 15, 20, 25, 30, 35, and 40 L min$^{-1}$. For the 125 μm sample the rheology experiments were performed at 0, 1, 2, 3, 4, 5, 7, 8, 9, and 10 L min$^{-1}$. For the 63 μm sample the rheology experiments were performed at 0, 0.25, 0.5, 1, 1.5, 2, 3, 4, and 5 L min$^{-1}$.

## Particle volume fraction

To calculate the bulk density, $\rho_b$, and particle volume fraction, $\phi$ we require an approximation of the volume occupied by the sample in the powder cell at different stages of fluidization. The volume of ash, $V$, in the powder cell was approximated using the measured height of the sample at a range of volumetric gas flow rates. The volume occupied by measuring geometry (Fig. S5) was considered such that the reported volumes are of the sample only (Fig. S6). Sample heights were only measured at a limited number of volumetric flow rates (Table S1), and we therefore interpolate the height occupied by the sample at other flow rate values, assuming a linear relationship between flow rate and height between the measured values (Fig. S7). The measured heights were obtained by contemporary observations of the fluidization experiments at different flow rates.

The bulk density $\rho_b$ and particle volume fraction $\phi$ were then calculated for all measurement points as follows:

$$\rho_b = m/V \tag{6}$$

$$\phi = \rho_b/\rho_s \tag{7}$$

where $m$ is sample mass loaded into the powder flow cell and $\rho_s$ is particle density. All these calculated values are provided in Table S1.

## Calculation of dimensionless groups

Here we summarise the parameters used to calculate the dimensionless numbers (Eqs. 1 to 3). Each sample is characterised by a representative particle diameter $d_p$, which was taken to be the median grain size $d_{50}$ – the value at which 50% of the sample volume is finer. We also examined the effect of this choice by also considering $d_{10}$ and $d_{60}$ as our representative particle size (used to produce the error bars in Fig. 2a). In our experiments, the drag coefficient was not measured, and we therefore explored a range of appropriate values. For spherical particles with high particle Reynolds numbers, the drag coefficient is constant[45] and approximately 0.4. However, for lower Reynolds number flows the drag coefficient is higher. Here we considered a drag coefficient in the range of 0.4–10, representing the range found in the literature[47]. The experimental points shown in Fig. 2a were calculated with an average drag coefficient estimate of $C_d = 5.2$.

The interstitial or carrier fluid (i.e., air) density, $\rho_f$ and viscosity, $\eta_f$ were assumed to be 1.225 kg m$^{-3}$ and 0.000018 Pa s respectively. All particles in all experiments were assumed to have an average sphericity value, $\chi$ of 0.7 which is in line with direct measurements on similar pyroclastic material[43]. We approximated the total stress $\sigma$ at the base of the sample, due to the weight of the bed as: $\sigma = \rho_b gh$, where $g$ = 9.81 m s$^{-2}$ is the acceleration due to gravity and $h$ is the bed height.

The mixture permeability was estimated using Darcy's law of fluid flow through a permeable bed[43]:

$$k = -\frac{Q\eta_f h}{A\Delta P} \qquad (8)$$

where $Q$ is the volumetric flow rate of fluid into the bed, $A$ is the cross-sectional area of the bed, and $\Delta P$ is the pressure drop across the bed. The area of the bed is calculated directly from the geometry of the fluidization cell (Fig. S5). The pressure drop and flow rate were measured directly in the experiments, and we calculated the Darcy permeability using Eq. 8, valid for laminar flows. The inherent relationship between the Darcy permeability and bed porosity is shown in Figure S8. At lower bed porosities, or higher bulk densities (i.e., for lower fluid fluxes and pressure drops), the permeability values are widely dispersed across several orders of magnitude. At higher porosities and lower bulk densities, there is an approximately linear increase in permeability with porosity for all grain size samples except the coarsest sample.

### Calculating the minimum fluidization velocity

The minimum fluidization velocity, $u_f$ can be predicted based on the particle characteristics alone when the exact void fraction of the bed and the particle concentration at the point of fluidization are not well constrained. Here, we wanted to independently calculate $u_f$ to facilitate comparison to our experimental observations in Fig. 1. We therefore applied the following, widely used[44,66], relationship:

$$u_f = \frac{\eta_f}{\rho_f d_p}\left(\sqrt{(33.7^2 + 0.0408\text{Ar})} - 33.7\right) \qquad (9)$$

Where $d_p$ is the representative particle diameter, here taken to be the median, $d_{50}$. Ar is the Archimedes number (a ratio of gravity to viscous forces) and given by:

$$\text{Ar} = \frac{\rho_f g(\rho_s - \rho_f)d_p^3}{\eta_f^2} \qquad (10)$$

Finally, $u_f$ was converted to $Q_f$ using $Q_f = Au_f$.

### Rheological model fitting

Our experimental datasets show that many of the variably fluidized granular mixtures exhibit both shear-thinning and shear-thickening rheological behaviors with a yield stress (Figs. 3 and 4). To model this non-Newtonian behavior, we use a modified form of the Herschel−Bulkley equation, inspired by the work of Brown et al.[54]. This is Eq. 4 in the main text. The three terms, $\tau_y$, $a_1\dot{\gamma}^\alpha$ and $a_2\dot{\gamma}^\beta$ represent the yield stress of the granular mixture, the shear-thinning regime, and the shear-thickening regime, respectively. For a small number of datasets where a further transition to a second shear-thinning regime was observed, this region was not modeled as it contained too few data points. For these experiments, Table S2 shows the shear stresses above which the data were not included in the fitting. Equation 4 was fitted to the rheology datasets for each grain size and for each volumetric flow rate. Fitting was performed using the Matlab functions $fit$ − to fit a polynomial curve to the data, and $fittype$−to define the equation to fit. The yield stress and shear-

thinning regime were fitted together with the equation $\tau = \tau_y + a_1\dot{\gamma}^\alpha$, whilst the shear-thickening region was fitted with the curve $\tau = a_2\dot{\gamma}^\beta$. The point of transition from shear-thinning to shear-thickening behavior was selected as the minimum of the rheological curve, where the viscosity gradient transitions from negative to positive. If present, the second transition back to shear-thinning behavior was defined by the curve maximum.

The goodness of fit of Eq. 4 to each flow curve was determined by a coefficient of determination ($R^2$), and each fitting parameter is reported with 95% confidence bounds (determined using Matlab's $confint$ function). For some datasets that were not fluidized (500 μm at $Q$ = 0, 15, 30 L min$^{-1}$; 250 μm at $Q$ = 0, 10 L min$^{-1}$; 125 μm at $Q$ = 0, 1, 2 L min$^{-1}$; 63 μm at $Q$ = 0, 0.25, 0.5, 1 L min$^{-1}$), Eq. 4 could not be accurately fitted. However, these data clearly exhibit a yield stress—a key rheological parameter that can be compared against the rest of the data in our analysis.

### Reporting summary

Further information on research design is available in the Nature Portfolio Reporting Summary linked to this article.

## Data availability

All data generated during this study are included in the Supplementary Information and the Source Data files.

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

## Acknowledgements

This work was supported by NERC grants NE/W003767/1 and NE/W006286/1 awarded to TJJ. JD and TJJ were also supported by a Royal Society Exchange grant (IES\R2\212037). CC was supported by UKRI grant (MR/S035141/1). The authors thank Alex Wilson for collecting the pumice samples (with landowner permission) used in this study.

## Author contributions
The project was conceived by TJJ. The material characterisation was performed by TJJ. The rheology experiments were performed by AS. Formal data analysis and dimensional scaling was completed by TJJ and CC. All authors contributed to data interpretation. All authors (TJJ, AS, CC, JD, HMG) contributed to the writing, reviewing, and editing of the paper after an initial draft was prepared by TJJ.

## Competing interests
At the time of submission and publication, Abhishek Shetty was an employee of Anton Paar GmbH, the company that commercially markets the rheometer used in this study. The other authors have no competing interests to disclose.
