## [Peer Review File · Nature Communications]

Identifying rheological regimes within pyroclastic density currentsREVIEWER COMMENTS

Reviewer #1 (Remarks to the Author):

I have read the paper with interest. It could potentially represent a very original contribution to better explain the rheological behavior of PDC. My main concern is that the authors are trying to extend the obtained result to all types of PDC, but to me, they are more likely applicable to pumice flow with dilute-intermediate particle concentration only (at the lower limit between dilute and concentrated PDC). Below are some main points, but I consider important that the authors state the limitation of the result obtained here. In the annotated pdf, authors can find some additional comments.

1) The particles used for the experiment are pumices; PDC can span from pumice-rich (ignimbrite) to dense lava fragment (block-and-ash flow)

2) Experiments are conducted with variable particle volume fraction between 0.29 and 0.13; but based on previous work (i.e., Lube et al., 2020), this value corresponds to PDC at the transitional boundary between dilute and concentrated PDC (not to intermediate to concentrated PDC which limit is considered for volume concentration > 0.3). In this sense, the proposed model should be restricted to this PDC category.

3) Scaling. The range of the Stokes value here considered for real PDC, between 1 and 10000, has been described for PDC in a concentrated regime, so, according to the previous comment, I'm not sure that it is a valid assumption. Please discuss.

Line 244-250 These observations are not conclusive enough to support the model presented here; they can have different interpretations.

Reviewer #2 (Remarks to the Author):

Review of: Identifying rheological regimes within pyroclastic density currents

Authors: Jones, Shetty, Chalk, Dufek, and Gonnermann

Reviewer: Benjamin J. Andrews, Smithsonian Global Volcanism Program

This manuscript presents a model for the rheology of pyroclastic density currents (PDCs). This model is based upon experimental results and describes a complex non-Newtonian rheology that can exert strong influences on PDC behavior including runout distance, ability to channelize, formation of dilute clouds, and generation of coignimbrite plumes. The authors show that different aspects of the rheology (e.g., suspension jams and deposition, shear-thinning, and shear-thickening) can be predicted as a function of shear stress and a normalized volumetric flow rate (this is presented as a schematic regime diagram). The results of this work can be incorporated into numerical models for more accurate forecasts of PDC behavior.

The authors demonstrate the rheology of particle-gas mixtures can be predicted as a function of shear stress and fluidization. Importantly, they show that these granular mixtures have: 1) a yield stress, 2) a region of shear-thinning, and 3) a region of shear-thickening. The details of how rheology varies can be placed into the context of natural concentrated to intermediate PDCs to predict deposition and aggradation. The yield strength depends on particle size, such that regions of the current with finer particles should have lower yield stresses.

This work is a significant advancement in our understanding of the rheology of concentrated to intermediate density PDCs. The results presented here can help to better quantify our understanding of PDC behavior and, hopefully, be incorporated into future numerical models for better forecasts of PDC runout and deposition. This is an important contribution for basic science and for hazard mitigation.

There are only a few areas where the paper need improvement. First, the paper should be clear that it is primarily applicable to concentrated to intermediate PDCs (this is stated at about line 314, but could be emphasized at a few points earlier in the text). Second, and related to the first point, the paper should describe how “dilute” the PDC can be for the effects of yield stress on deposition to still matter (see about lines 244-255).

I recommend acceptance with very minor revisions. This is a very well-written paper that discusses important processes in the dynamics and deposition of pyroclastic density currents. The results have implications for hazard mitigation as well as applicability in fields outside of volcanology. Detailed comments are presented below.

Benjamin J. Andrews
Smithsonian Global Volcanism Program

Line-by-line comments

35 – change to “eruptions” (plural)

37 – change to “and comprise a multiphase...”

38 – change to “many kilometers” – I don’t mean to nitpick, but, to me, “several” suggests probably less than 10 km, whereas “many” could be 10s of more.

40 – suggest adding “length-scales and timescales, ...”

41 – suggest changing “hotly debated” to “intensely studied”

57 – “Consequently, this limits the ability of models to accurately...”

78 – I would emphasize here that these measurements were also of the deposit, which is not the same thing as the PDC from which it came.

82 – rephrase as “We use a powder flow cell...”

105 – change to “varied slightly”

119 – the phrasing “2-blade stirrer geometry” is awkward. Should this be a “2-blade stirrer”?

159 – it seems like this paragraph should be qualified with “Dense PDCs in nature...” The paragraph is well-written, and makes sense, but it is very much focused on the dense (into intermediate density) PDCs. I don’t know that the granular Stokes number is really applicable to a dilute PDC.

209 – suggest rephrase as “... gas-pyroclast suspension depends on the imposed shear rate.”

210 – shouldn’t this be “all pyroclast suspensions... exhibit a non-Newtonian rheology”? You are

measuring suspensions, not the internal rheologies of the individual rocks.

217 – suggest changing “larger” to “higher”

219 – insert a comma after “given grain size”

242 – suggest deleting the final sentence of this paragraph (“The implications will now be explored.”)

244-255 – I like this paragraph and the way that you put the yield stress into the context of deposition and aggradation. But there needs to be a little better explanation of how this can be extended into currents that are becoming increasingly dilute, or what you mean by “dilute.” It seems that a truly dilute current (e.g., one with little to no particle-particle interactions) would not have a yield stress. Please clarify.

276 – you might be able to delete the comma at the start of this line

295 – suggest deleting “ignimbrite” as this can be a somewhat loaded term. (everything you are describing here can be applied to the deposits of a block-and-ash flow, but BAF deposits are not generally called ignimbrite deposits)

297-300 – I see where you are going here, but I would rewrite these two sentences. Right now, it kind of reads like “decoupling is favored when decoupling can occur” (your point is good, it just needs to be rephrased)

309 – rephrase as “Whereas we intentionally focused...” (this gets rid of the word “here” so that you can use it at the start of the next paragraph)

403 – delete the word “again”

Reviewer #3 (Remarks to the Author):

This study presents rheological measurements on fluidized pyroclastic materials with the aim of defining a rheological model that can be included in numerical models to simulate the emplacement of pyroclastic density currents. The subject is of interest to the volcanological community, but the manuscript presented is confusing in several respects and the implications for natural systems are far from obvious.

The choice of material used for rheological measurements is surprising given that the authors aim to define a rheological law for pyroclastic density currents. Why use crushed pumices from fall deposits, which are angular, rather than particles from a flow deposit, which are known to be more rounded? The shape of the particles is important, particularly at high particle concentrations, as it partly controls particle interactions and therefore the apparent viscosity measured. Furthermore, what is the behavior of a polydisperse mixture typical of a PDC? I can see the advantage of making measurements with sieved fractions to understand the effect of particle size, but in the end it is very difficult to appreciate the rheological behavior of a PDC from the results presented. Finally, it is important to note that the measurements were made at room temperature, at which fine pyroclastic materials (in this case less than 63 μm) are known to be cohesive. This certainly has an influence on the results, particularly the yield stress discussed in the manuscript.

The fluidization behavior of the gas-pyroclast mixtures shown in Fig. 1 is not rigorously characterized. The plateau of maximum pressure drop at increasing flow rate is clearly not reached (except for the largest particles), contrary to what is stated (line 115). In this context, how did the authors define the critical flow rate Q_f ? This is a crucial point, as Q_f is used to normalize the flow rate in Figs. 1b and 4c. Therefore, how much confidence can we place in the rheological map presented in Fig. 4c? Furthermore, what fraction of the bed weight is compensated by the fluid pressure at ΔP_{\max} corresponding to Q_{\max} ? To determine the degree of fluidization, the ratio $\Delta P / (\rho_{\text{bulk}} * g * h)$ should be considered. Another point concerns the behavior of the fluidized bed: is it homogeneous or heterogeneous (i.e., with gas bubbles)?

Discussing dynamic similarity is interesting, but several sentences in this section cast doubt on the physical regimes considered by the authors. For example, it is stated that viscous drag is significant at low Stokes number and results in high pore pressure (lines 164-165), which contradicts the sentence in lines 171-172 which states that the lowest St values correspond to experiments with no gas flux (i.e., no gas flux implies no pore pressure).

Yield stress is widely discussed by the authors, but several points are confusing. First, the origin of the yield stress observed is unclear and needs to be considered. The authors mention that yield stress is measured in experiments at $2xQ_f$, in which the material is highly expanded. This could suggest an experimental artifact due to the design of the device, possibly particle friction on the smooth edges of the shear cell. Secondly, yield stress is certainly not the cause of reverse grading and levee-lobe morphology (lines 244-245), which are explained in numerous publications by kinetic sieving and particle-size segregation in polydisperse flows.

The attempt to define a regime diagram is welcome, but the way the analysis is presented is confusing. Not surprisingly, equation 4 fits the data well (line 278), as the fitting factors are derived empirically (line 276). Also, it is stated that the critical shear rate γ_{c1} corresponds to the local minima $d_{\tau} / d_{\dot{\gamma}} = 0$, which is apparently never the case. The regime boundaries in Figures 4c-d are poorly constrained (please see comment on Q_f above) and the implications for PDCs are speculative without further justification. Please note that in Figure 4d the particle concentration increases with gas flow, which makes no sense.

L64. The seminal study of Wilson (ref. 42) should be cited instead.

L193. from d10-d60, where the number

L596. pyroclastic flows

Point-by-point reply.

Reviewer comments are in standard font. Our responses are listed in *blue italics*. We also provide a full manuscript with 'tracked changes'.

Reviewer #1 (Remarks to the Author):

I have read the paper with interest. It could potentially represent a very original contribution to better explain the rheological behavior of PDC. My main concern is that the authors are trying to extend the obtained result to all types of PDC, but to me, they are more likely applicable to pumice flow with dilute-intermediate particle concentration only (at the lower limit between dilute and concentrated PDC). Below are some main points, but I consider important that the authors state the limitation of the result obtained here. In the annotated pdf, authors can find some additional comments.

Many thanks for these useful edits. We have made the changes, and we expand further on your key points below.

1) The particles used for the experiment are pumices; PDC can span from pumice-rich (ignimbrite) to dense lava fragment (block-and-ash flow)

We have now made this clear from the outset. Text added to the abstract (line 25) and the introduction (line 84). It is also made clear that our experiments are limited to pumice rich.

2) Experiments are conducted with variable particle volume fraction between 0.29 and 0.13; but based on previous work (i.e., Lube et al., 2020), this value corresponds to PDC at the transitional boundary between dilute and concentrated PDC (not to intermediate to concentrated PDC which limit is considered for volume concentration > 0.3). In this sense, the proposed model should be restricted to this PDC category.

The boundaries delineated by Lube et al., (2020) Ref 1 are approximate and are sensitive to the grain size distribution, which in turn sets the critical volume fraction or maximum packing. In the classification of Lube et al. (2020) intermediate regime behaviour is described by the onset of meso-scale clustering and the concentrated regime considered the point at which clustering is inhibited to close packing. Lube et al (2020) based on a range of experiments give the onset of clustering between $\sim 10^{-3}$ to 10^{-2} . In this characterization, this is the boundary between dilute and intermediate regimes. Clustering is inhibited by $\sim 3-4 \times 10^{-1}$. So, in this dynamics-based characterization of flow regimes, we would still be operating in the intermediate to concentrated regimes.

In our situation maximum packing occurs at about this volume fraction as well, i.e., ~ 0.29 . It is important to note that the Lube et al., transitions are approximate and based on particle characteristics and we have hence added and edited the following text:

Line 83 onwards:

Here, we perform a series of scaled laboratory experiments to directly image the internal dynamics and rheological properties of pumice-rich, intermediate to concentrated PDCs, with particle volume fractions, ϕ between 0.29 (corresponding to the maximum packing fraction in the unfluidized state, ϕ_m) and 0.13 (when fluidized). Here we use the scaling of Lube et al. to describe these regimes and emphasize that the transitions between regimes are approximate and based on particle characteristics. Thus, even at the lowest ϕ of 0.13, the suspension still falls within the intermediate regime with evident particle clustering.

3)Scaling. The range of the Stokes value here considered for real PDC, between 1 and 10000, has

been described for PDC in a concentrated regime, so, according to the previous comment, I'm not sure that it is a valid assumption. Please discuss.

This range in granular Stokes number represents concentrated to intermediate PDCs, as values less than 1 lie in the viscous regime and are more typical of dilute PDCs (Dufek, 2016). As discussed in the previous comment, it is challenging to differentiate precise boundaries between the intermediate and concentrated regimes. The lower end of the Stokes number range has been extended to 1 to include the full range of intermediate PDCs (it was previously 5, taken from Lube et al Fig. 4). Figure 2 and the associated text have been updated to clarify that this range represents intermediate to concentrated regimes, and not only concentrated regimes.

Line 244-250 These observations are not conclusive enough to support the model presented here; they can have different interpretations.

We agree that some of the observations presented can have multiple explanations, one of which is the presence of a yield stress. This has been made clearer in the text. We have also added additional statements to support our points.

Reviewer #2 (Remarks to the Author):

This manuscript presents a model for the rheology of pyroclastic density currents (PDCs). This model is based upon experimental results and describes a complex non-Newtonian rheology that can exert strong influences on PDC behavior including runout distance, ability to channelize, formation of dilute clouds, and generation of coignimbrite plumes. The authors show that different aspects of the rheology (e.g., suspension jams and deposition, shear-thinning, and shear-thickening) can be predicted as a function of shear stress and a normalized volumetric flow rate (this is presented as a schematic regime diagram). The results of this work can be incorporated into numerical models for more accurate forecasts of PDC behavior.

The authors demonstrate the rheology of particle-gas mixtures can be predicted as a function of shear stress and fluidization. Importantly, they show that these granular mixtures have: 1) a yield stress, 2) a region of shear-thinning, and 3) a region of shear-thickening. The details of how rheology varies can be placed into the context of natural concentrated to intermediate PDCs to predict deposition and aggradation. The yield strength depends on particle size, such that regions of the current with finer particles should have lower yield stresses.

This work is a significant advancement in our understanding of the rheology of concentrated to intermediate density PDCs. The results presented here can help to better quantify our understanding of PDC behavior and, hopefully, be incorporated into future numerical models for better forecasts of PDC runout and deposition. This is an important contribution for basic science and for hazard mitigation.

Thank you for your positive and constructive review.

There are only a few areas where the paper need improvement. First, the paper should be clear that it is primarily applicable to concentrated to intermediate PDCs (this is stated at about line 314, but could be emphasized at a few points earlier in the text).

Thank you. We have now made this clear from the outset. Text added to the abstract (line 25) and the introduction (lines 53 and 84).

Second, and related to the first point, the paper should describe how “dilute” the PDC can be for the effects of yield stress on deposition to still matter (see about lines 244-255).

We have now made it clearer in the introduction that we only focus on concentrated and intermediate flows (as per comment above). In our comments around deposition, we have added the statement: “This would occur up until a dilute concentration limit where particle-particle interactions

and any corresponding yield strength is negligible". We have not provided an exact concentration limit as this will likely depend on particle size, shape, etc.

I recommend acceptance with very minor revisions. This is a very well-written paper that discusses important processes in the dynamics and deposition of pyroclastic density currents. The results have implications for hazard mitigation as well as applicability in fields outside of volcanology. Detailed comments are presented below.

Benjamin J. Andrews
Smithsonian Global Volcanism Program

Line-by-line comments

35 – change to “eruptions” (plural)

37 – change to “and comprise a multiphase...”

38 – change to “many kilometers” – I don’t mean to nitpick, but, to me, “several” suggests probably less than 10 km, whereas “many” could be 10s of more.

40 – suggest adding “length-scales and timescales, ...”

41 – suggest changing “hotly debated” to “intensely studied”

57 – “Consequently, this limits the ability of models to accurately...”

78 – I would emphasize here that these measurements were also of the deposit, which is not the same thing as the PDC from which it came.

82 – rephrase as “We use a powder flow cell...”

105 – change to “varied slightly”

119 – the phrasing “2-blade stirrer geometry” is awkward. Should this be a “2-blade stirrer”?

159 – it seems like this paragraph should be qualified with “Dense PDCs in nature...” The paragraph is well-written, and makes sense, but it is very much focused on the dense (into intermediate density) PDCs. I don’t know that the granular Stokes number is really applicable to a dilute PDC.

209 – suggest rephrase as “... gas-pyroclast suspension depends on the imposed shear rate.”

210 – shouldn’t this be “all pyroclast suspensions... exhibit a non-Newtonian rheology”? You are measuring suspensions, not the internal rheologies of the individual rocks.

217 – suggest changing “larger” to “higher”

219 – insert a comma after “given grain size”

242 – suggest deleting the final sentence of this paragraph (“The implications will now be explored.”)

244-255 – I like this paragraph and the way that you put the yield stress into the context of deposition and aggradation. But there needs to be a little better explanation of how this can be extended into currents that are becoming increasingly dilute, or what you mean by “dilute.” It seems that a truly dilute current (e.g., one with little to no particle-particle interactions) would not have a yield stress. Please clarify.

276 – you might be able to delete the comma at the start of this line

295 – suggest deleting “ignimbrite” as this can be a somewhat loaded term. (everything you are describing here can be applied to the deposits of a block-and-ash flow, but BAF deposits are not generally called ignimbrite deposits)

297-300 – I see where you are going here, but I would rewrite these two sentences. Right now, it kind of reads like “decoupling is favored when decoupling can occur” (your point is good, it just needs to be rephrased)

309 – rephrase as “Whereas we intentionally focused...” (this gets rid of the word “here” so that you can use it at the start of the next paragraph)

403 – delete the word “again”

All these changes have been made.

Reviewer #3 (Remarks to the Author):

This study presents rheological measurements on fluidized pyroclastic materials with the aim of defining a rheological model that can be included in numerical models to simulate the emplacement of pyroclastic density currents. The subject is of interest to the volcanological community, but the manuscript presented is confusing in several respects and the implications for natural systems are far from obvious.

The choice of material used for rheological measurements is surprising given that the authors aim to define a rheological law for pyroclastic density currents. Why use crushed pumices from fall deposits, which are angular, rather than particles from a flow deposit, which are known to be more rounded? The shape of the particles is important, particularly at high particle concentrations, as it partly controls particle interactions and therefore the apparent viscosity measured.

We agree that some particles in the PDCs are rounded however this is not ubiquitous across all size fractions. Rounding rather changes as a function of particle size. Here we focused on changing the fluidization state (i.e., gas flux) and the particle size in order to conduct the experiments in a systematic way such that the key controlling factors could be identified. We agree that another study in the future looking at the effect of particle shape would be useful. However, we still expect to observe the different rheological regimes – the main point of this paper—in experiments with rounded particles. Only the precise value of the yield strength, for example, will vary as it does for different particle sizes and concentrations.

Furthermore, what is the behavior of a polydisperse mixture typical of a PDC? I can see the advantage of making measurements with sieved fractions to understand the effect of particle size, but in the end it is very difficult to appreciate the rheological behavior of a PDC from the results presented.

We agree that this would be interesting, however is beyond the scope of this study. As above, we would still expect to observe the different rheological regimes and their transitions – the main point of this paper.

Finally, it is important to note that the measurements were made at room temperature, at which fine pyroclastic materials (in this case less than 63 μm) are known to be cohesive. This certainly has an influence on the results, particularly the yield stress discussed in the manuscript.

This is noted in the manuscript (line 420), and we agree with you. This is why we have not performed any experiments with particles less than 63 μm .

The fluidization behavior of the gas-pyroclast mixtures shown in Fig. 1 is not rigorously characterized. The plateau of maximum pressure drop at increasing flow rate is clearly not reached (except for the largest particles), contrary to what is stated (line 115).

Thank you for encouraging us to expand on this more. Firstly, all experiments did reach a plateau of maximum pressure drop. This is just a matter of graph visualization. In Figure 1a, the maximum volumetric flow rate used in the experiments varies over an order of magnitude based on the grain size. To account for this, in Figure 1b, we normalise this axis to the max flow rate used, and in that figure, it is clearer that all experiments reached a plateau. Clarity has been added to the main text and the figure caption.

In this context, how did the authors define the critical flow rate Q_f ? This is a crucial point, as Q_f is used to normalize the flow rate in Figs. 1b and 4c. Therefore, how much confidence can we place in the rheological map presented in Fig. 4c? Furthermore, what fraction of the bed weight is

compensated by the fluid pressure at ΔP_{\max} corresponding to Q_{\max} ? To determine the degree of fluidization, the ratio $\Delta P/(\rho_{\text{bulk}}*g*h)$ should be considered. Another point concerns the behavior of the fluidized bed: is it homogeneous or heterogeneous (i.e., with gas bubbles)?

Excellent point, thank you. We have now theoretically calculated Q_f and shown these values as vertical dashed lines in Figure 1. As you can see, they are close to the plateau onset recorded by the experiments. We prefer this approach rather than estimating the bed weight, because our height measurements (thus bulk bed density) are not continuous and were performed only 4 times per experiment. Interpolation in our case would lead to unnecessary and unquantifiable errors. The calculation of Q_f is now detailed in a separate methods section (line 493), outlined in the main text (line 126) and the data are clearly presented in Figure 1. A comment about the bed dynamics is added on line 120. The rheological maps (e.g., Figure 4c) have now been updated to include the calculated Q_f as suggested.

Discussing dynamic similarity is interesting, but several sentences in this section cast doubt on the physical regimes considered by the authors. For example, it is stated that viscous drag is significant at low Stokes number and results in high pore pressure (lines 164-165), which contradicts the sentence in lines 171-172 which states that the lowest St values correspond to experiments with no gas flux (i.e., no gas flux implies no pore pressure).

The development of elevated pore pressure typically requires 1. a gas source (or reduction of pore volume) and 2. Relative reduction of permeability. In natural flows, gas sources (1) might be provided by entrainment, but in these experiments are provided by the external gas flux. The granular Stokes number here describes the relative importance of viscous interactions and gravity in concentrated granular flows, with lower granular Stokes numbers indicating a greater influence of viscous coupling with the fluid much as you would assume with lower permeability. While lower permeability may help with the accumulation of pore pressure, low granular Stokes numbers are not sufficient to establish high pore pressure (in this case the lack of a gas source limits pore pressure development). We have tried to clarify this in the text:

Lines 172-173: St_g decreases with both increases in the interstitial fluid viscosity and with reduction in permeability.

Lines 189-192: The lowest St_g values correspond to the experiments with no gas flux ($Q = 0 \text{ L min}^{-1}$), which have a significantly lower permeability than the other experiments, but we note that pore pressure here remains low due the lack of an additional gas source

Yield stress is widely discussed by the authors, but several points are confusing. First, the origin of the yield stress observed is unclear and needs to be considered. The authors mention that yield stress is measured in experiments at $2xQ_f$, in which the material is highly expanded. This could suggest an experimental artifact due to the design of the device, possibly particle friction on the smooth edges of the shear cell.

Thank you for this, the yield stress origin is now mentioned on line 258. We disagree that there is any experimental artifact. Even in our experiments with the highest gas flow rate and where the particle fraction, ϕ is the lowest (0.13), the particle concentrations are close to the maximum packing fraction ϕ_m . In the most dilute case, $\phi/\phi_m = 0.13/0.29 = 0.45$, this is still sufficient for particle-particle interactions and thus a yield stress.

Secondly, yield stress is certainly not the cause of reverse grading and levee-lobe morphology (lines 244-245), which are explained in numerous publications by kinetic sieving and particle-size segregation in polydisperse flows.

While sorting and grain size variations can amplify the impact of levee formation and rheology, there have also been several experiments that show levee formation without the presences of any grain

size sorting. In particular, we refer the reader to the work of Rocha et al., 2019 (and references therein) that show levee formation in dry granular, monodisperse flows. In particular, Rocha et al. (2019) states:

“Both of these sets of experiments are dry and have a very narrow range of particle sizes, but, as Félix & Thomas (2004) showed, static levees still form. This suggests that neither interstitial fluid nor particle-size segregation are essential to the self-channelisation process, although they may strongly enhance its effects (Pouliquen, Delour & Savage 1997; Pouliquen & Vallance 1999; Félix & Thomas 2004; Goujon, Dalloz-Dubrujeaud & Thomas 2007; Iverson et al. 2010; Woodhouse et al. 2012; Kokelaar et al. 2014; Baker, Johnson & Gray 2016b).”

Rocha, F.M., Johnson, C.G. and Gray, J.M.N.T., 2019. Self-channelisation and levee formation in monodisperse granular flows. *Journal of Fluid Mechanics*, 876, pp.591-641.

We have added the following lines to the text to clarify:

Lines 268-271: While changes in grain sizes due to sorting may further accentuate these feature (in particular levee formation), previous work on dry granular flows with narrow grain size distributions have still exhibited self-channelization and levee formation owing to changes in rheology

The attempt to define a regime diagram is welcome, but the way the analysis is presented is confusing. Not surprisingly, equation 4 fits the data well (line 278), as the fitting factors are derived empirically (line 276). Also, it is stated that the critical shear rate γ_{c1} corresponds to the local minima $d_{\tau}/d_{\dot{\gamma}}=0$, which is apparently never the case.

It is the standard procedure across the entire rheological community to fit these parameters in the Hershel Buckley model. They are indeed empirical fitting constants.

The critical shear rate corresponds to the local minima of the viscosity-strain rate curve, not the stress-strain rate curve. This typo has now been corrected in the text on line 306, and we thank the reviewer for pointing this out. Mathematically, the minima of the curve is when the gradient is zero – i.e. when the gradient shifts from being negative to positive.

The regime boundaries in Figures 4c-d are poorly constrained (please see comment on Q_f above) and the implications for PDCs are speculative without further justification. Please note that in Figure 4d the particle concentration increases with gas flow, which makes no sense.

We have adjusted the regime diagram Figure 4, based on the Q_f recommendation (see above). We agree that we give possible implications, this is now stated clearly in the text. The schematic (Fig 4d) has also been changed to avoid giving the impression that there is a yield strength at infinite gas flow rate. We agree that particle concentration does not increase with gas flow rate, this was never our intention and we have removed the misleading arrows in the axis labels. Thank you for spotting this.

L64. The seminal study of Wilson (ref. 42) should be cited instead.

L193. from d_{10} - d_{60} , where the number

L596. pyroclastic flows

All these changes have been made.

REVIEWERS' COMMENTS

Reviewer #1 (Remarks to the Author):

I'm very happy with all changes made, as well as to the answers provided to my comments. I consider that the paper represents a very original contribution and it will be a reference for the scientific community working on PDC.

Reviewer #3 (Remarks to the Author):

The authors have done a good job of reviewing and clarifying the points I raised. Their answers on the particles used, the pressure plateau, the Stokes number, the cause of a yield stress and its effect on levee-lobe morphology (based on recent references) are convincing. It is an interesting study.

Point-by-point reply.

Reviewer comments are in standard font. Our responses are listed in *blue italics*. We also provide a full manuscript with 'tracked changes'.

Reviewer #1 (Remarks to the Author):

I'm very happy with all changes made, as well as to the answers provided to my comments. I consider that the paper represents a very original contribution and it will be a reference for the scientific community working on PDC.

Many thanks for helpful and constructive review.

Reviewer #3 (Remarks to the Author):

The authors have done a good job of reviewing and clarifying the points I raised. Their answers on the particles used, the pressure plateau, the Stokes number, the cause of a yield stress and its effect on levee-lobe morphology (based on recent references) are convincing. It is an interesting study.

Many thanks for helpful and constructive review.